# Unmet Needs of Australians in Endometriosis Research: A Qualitative Study of Research Priorities, Drivers, and Barriers to Participation in People with Endometriosis

**DOI:** 10.3390/medicina59091655

**Published:** 2023-09-13

**Authors:** Nora Giese, Emilee Gilbert, Alexandra Hawkey, Mike Armour

**Affiliations:** 1Praxis für Chinesische Medizin, 53113 Bonn, Germany; info@praxis-giese.com; 2School of Psychology, Western Sydney University, Penrith, NSW 2751, Australia; e.gilbert@westernsydney.edu.au; 3Translational Health Research Institute (THRI), Western Sydney University, Penrith, NSW 2751, Australia; a.hawkey@westernsydney.edu.au; 4NICM Health Research Institute, Western Sydney University, Penrith, NSW 2751, Australia; 5Medical Research Institute of New Zealand (MRINZ), Wellington 6021, New Zealand

**Keywords:** endometriosis, research priorities, focus groups, unmet needs, Australia

## Abstract

*Background and Objectives*: Endometriosis causes significant personal and societal burden. Despite this, research funding lags behind other chronic conditions. Determining where to prioritise these limited funds is therefore vital. Research priorities may also differ between individuals with endometriosis and clinicians/researchers. The aim of this research project is to explore research priorities and factors shaping participation in endometriosis research from the perspective of people with endometriosis in Australia. *Materials and Methods*: Four focus groups involving 30 people with endometriosis were conducted and analysed using qualitative inductive content analysis. *Results*: Two categories were developed from the data: unmet research needs and motivators and barriers to participation in endometriosis research. Participants expressed interest in developing non-invasive diagnostic tools and a more multidisciplinary or holistic approach to treatment. Participants urgently desired research on treatment options for symptom management, with many prioritising non-hormonal treatments, including medicinal cannabis and complementary medicine. Others prioritised research on the causes of endometriosis over research on treatments to assist with prevention and eventual cure of the disease. The main drivers for participating in endometriosis research were hope for symptom improvement and a reduction in time to diagnosis. Research design features that were important in supporting participation included ease of access to testing centres (e.g., for blood tests) and sharing test results and automated data collection reminders, with simple stra-tegies to record data measurements. Research incentives for younger people with endometriosis and a broad dissemination of information about research projects was considered likely to increase participant numbers. Barriers included time commitments, a lack of flexibility around research appointments for data collection, travel or work commitments, concerns about the safety of some products, and trying to conceive a child. *Conclusions*: People with endometriosis were open to participating in research they felt aligned with their needs, with a significant focus on diagnostic tools and symptom relief. However, researchers must co-design approaches to ensure convenience and flexibility for research participation.

## 1. Introduction

Endometriosis affects around one in nine women and those assigned female at birth in Australia by the age of 44 [1]. Due to the stigma and silencing surrounding the menstrual cycle [2,3] and difficulties with diagnosis [4], it is estimated that people in Australia can experience a diagnostic delay of between 6.4 and 8 years [5,6]. Endometriosis causes significant pain and fatigue [5] and can negatively impact all aspects of an individual’s life, including work, education, sexual and social relationships, self-identity, and body image [5,7,8]. It also has a significant cost of illness burden of over AUD 9.7 billion per year in Australia [9]. Despite the significant personal and societal burden, endometriosis research has been under-funded and under-researched compared to other chronic conditions with a similar prevalence and health-care burden [10]. This has led to a limited understanding of the disease aetiology and slowing of innovations in diagnosis and treatment [11,12].

In Australia, the National Action Plan for Endometriosis, launched in 2018, has three major goals: (i) awareness and education, (ii) clinical management and care, and (iii) research [13]. In order to meet goal three within the constraints of limited funding, the setting of research priorities is vital [14].

Endometriosis research priorities have been developed in the past [10,12,15,16,17,18]. The most comprehensive publication on endometriosis research recommendations reported the findings of the 3rd International Consensus Workshop on Research Priorities in Endometriosis in 2014 with 60 endometriosis investigators from 19 countries [17]. One hundred and seven research priorities were recommended, concerning all key areas of endometriosis research: pathogenesis and pathophysiology, symptoms, diagnosis, classification and prognosis, disease and symptom management, low-income countries and low-resource settings, and research policy. Although this compilation of endometriosis research topics provides a comprehensive overview of the aspects of endometriosis that are not well understood or require development, it remains unclear which of these 107 recommendations should be prioritised according to patients’ needs.

Previous research priority-setting efforts have mostly included researchers and/or clinicians, with only one reporting significant consumer involvement [10]. The importance of consumer involvement in health research is increasingly being understood [19]. Patient-centred medicine demands consumer involvement in health research, an issue debated and advocated for by bodies including the Cochrane collaboration, the Consumers Health Forum of Australia, and the UK’s National Institute of Health and Clinical Excellence [19,20]. Endometriosis research priorities identified by consumers often differ from those developed by clinicians and scientists in key areas [21]. For example, consumers and fa-mily members are more likely to prioritise education/awareness, emotional impact, and comorbid conditions, whereas healthcare professionals and scientists are more interested in cause/pathology or risk factors for endometriosis, diagnosis and screening, treatment, and fertility [21].

Another challenge for those with endometriosis is the significant pain and fatigue caused by the disease, as well as the unpredictable nature of the pain, known as “endo flares.” This can act as a barrier to consumer participation in research. This can be parti-cularly difficult when data collection utilises in-person measurements that are tied to particular timepoints. An unexpected endo flare or the arrival of the menstrual period can cause significant pain, potentially leading to substantial dropouts or missing data [22].

Considering the importance of consumer-led priority setting, this research aims to explore what people with endometriosis perceive to be unmet research needs and how endometriosis research can be tailored to meet consumers’ specific requirements. By prioritising the voices of those with endometriosis, we are better placed to understand which research topics should be prioritised and how research can be conducted so that people with endometriosis are empowered to take part in research that both interests and benefits them.

## 2. Materials and Methods

A qualitative design consisting of focus group interviews was used to explore the research needs of people with endometriosis and, in particular, future research priorities. Ethical approval was obtained from the Western Sydney University Human Ethics Committee (Approval H13131) in February 2019.

### 2.1. Participants

Participants were eligible for inclusion if they were aged over 18, lived in Australia at the time, and had a diagnosis of endometriosis via laparoscopy. Recruitment occurred via social media posts in February and March 2019. Australian endometriosis support organi-sations, including Endometriosis Australia and QENDO, posted the research invitation on their Facebook and Instagram pages, with a combined follower count of more than 45,000 people. Participants were reimbursed AUD 20 for their time via a gift card upon completion of the study.

A total of 30 people participated in the focus groups analysed as part of this research project. Participants were grouped in the age ranges of 18–24 (*n* = 7, 23%), 25–34 (*n* = 14, 47%), and ≥35 (*n* = 9, 30%) years.

Most participants identified as Caucasian (*n* = 24, 80%), were in heterosexual marriages (*n* = 19, 63%) and did not have children (*n* = 25, 83%). Over half worked in full-time employment (*n* = 18, 60%) and had a minimum of a university undergraduate degree. See Table 1 for further demographic data.

### 2.2. Procedure

Four focus groups with 6–9 people, totalling 30 people, with a diagnosis of endometriosis were conducted in March 2019, audio-recorded, and transcribed verbatim. Focus groups were conducted via the online platform Zoom and lasted 70–90 min. Given that priorities and barriers may depend on age, focus groups were divided into the following brackets: 18–24, 25–34, and ≥35 years old. Questions addressed the areas of endometriosis research that need further investigation, why they are important to participants, and barriers to participating in a research project on endometriosis. See Appendix A for the complete focus group schedule. This paper only focuses on the questions related to the priorities and barriers to research participation. Additional analysis will be published separately.

### 2.3. Analysis

A qualitative content analysis was used to ascertain patients’ views on research priorities. This method is a form of analysis that allows researchers to systematically transform large amounts of qualitative data into an organised and concise summary of key results [23].

For data analysis, an inductive approach was used following the two-phase, eight-step process described by Roller (2019) [24]. Phase 1 included data generation and coding. Familiarisation with the content of the transcripts was then gained through repeated rea-ding. Each of the four focus group transcripts formed one unit of analysis. Codes were key ideas or statements identified in the data. To foster reliability, they were independently identified by two researchers (N.G. and E.G.). The codes were discussed amongst the researchers until consensus was achieved. The final codes were applied to all four transcripts by author one (N.G.). Coding and data analysis were carried out manually. Statements within the transcripts were labelled with the corresponding code by using the Microsoft Word Comment function. Phase 2 consisted of data analysis with categorisation and interpretation (N.G., E.G., and M.A.): Key codes were listed, grouped together, and labelled as a thematic category in a separate Word document. The categories were discussed amongst all of the researchers until consensus was achieved. In the final step, interpretations and implications were drawn and discussed amongst all of the researchers.

To quantify the participants’ statements, word search was used to search for codes across participants accounts. This process involved checking each participant to see whether they had made a statement about a specific code. An Excel spreadsheet was used to either note each participant’s statement about the specific code or note whether the participant had not made a statement about that code. A count was then made of how many participants had made a statement about that specific code. Representative quotes from each code are presented below, followed by the participant pseudonym and age range.

## 3. Results

Two categories were developed from the data: (i) unmet research needs and (ii) motivators and barriers to participation in endometriosis research.

### 3.1. Unmet Research Needs: Diagnosis, Treatment, and Cure

Participants described three main research areas that require exploration in future research: diagnosis, treatment, and the development of a cure.

Eleven out of 30 participants described how receiving an endometriosis diagnosis is a problematic process. This is due in part to the prolonged time to diagnosis and the invasive nature of laparoscopic surgery, which has historically been required for a confirmed diagnosis. Participants reported that an endometriosis diagnosis “seems to always be the last [option]” (Caroline, ≥35), with all other possible physical and psychological causes ruled out prior to being offered a laparoscopy. As Pam (25–34) said, “I had a healthy appendix taken out, and I was told I was stressed.” Gloria (≥35) explained the frustration she experienced, saying, “If you present to a doctor or gyno with these sorts of complications and pains, to not just be put down as, ‘Oh it’s this, oh it’s that. Oh, we can’t tell unless you have a laparoscopy’.” For Wendy, a diagnosis of endometriosis only came after having been repeatedly mis-diagnosed with other conditions, including a urinary tract infection and gonorrhoea:


*“I think a bit more research into that [diagnosis of endometriosis] would be good for when a woman just says she’s got pelvic pain—I’ve just been told so many times: ‘Oh, you’ve got a UTI [urinary tract infection], you’ve got gonorrhoea, you’ve got all these things’, and then the tests comes back and they: ‘Oh, you don’t have any of that’. It’s hard.”*
(Wendy, ≥35)

Pam (25–34) explained that a diagnosis that allowed her to know that “you’re not insane, and all this isn’t just in your head, there’s actually something would be a really good place to start.” Early detection could ameliorate the feeling of uncertainty around the perceived legitimacy of symptoms prior to the diagnosis. For Lana (25–34), this uncertainty was around the outcome of surgery and the hope of receiving a diagnosis given the significant cost associated with such surgery.


*“I also remember leading into my first surgery that I was also really nervous that I was going to be outlaying all of these costs of the surgery and all of those associated things—only to not know, if I was able going to be getting a diagnosis of any sort. I remember waking up in surgery and asking: ‘Did they find anything?’ Because I would have felt guilty spending our money on all of that, only to find out that, no, there was nothing there.”*
(Lana, 25–34)

Participants felt that “early intervention is really key” (Jenna, 25–34), as it is likely to change the treatment trajectory, resulting in better health outcomes, including a reduction in emotional burden. As Jan described:


*“I didn’t get diagnosed ‘till I was 34, and that was after years of infertility and pregnancy loss. If I had known about this 10 years ago because someone did a blood test or a scan of some sort, it could have saved me all of that trauma.”*
(Jan, 25–34)

Participants stated that early detection could include genetic screening or the formal collection of early symptoms beyond pelvic pain alone. Gloria explained feeling not “listened to” when describing her endometriosis symptoms to healthcare professionals and thus saw value in a pre-laparoscopy test that may indicate endometriosis:


*“There is enough people—there’s one in ten women, so there’s a lot people presenting with all these conditions, there needs to be something that they can do, a test other than just the laparoscopy. If they could research into that so you could see that, yes, you are a candidate for endometriosis, so now let’s do a laparoscopy, but at least be listened to prior to just have the laparoscopy.”*
(Gloria, ≥35)

Twenty out of 30 participants described current endometriosis treatments as limited, with potentially effective treatment such as surgery not always having the desired results for pain reduction. Participants reported that their treatment outcomes were unsatisfying or even detrimental, telling us that it “made things worse” and makes you “feel like crap,” with symptoms being “worse after surgery” or that “[surgeries] didn’t work well” and “nothing seemed to have worked.” As some participants described, “even since the surgeries it’s still constant pain” (Phyllis, ≥35). A lack of treatment options was particularly apparent for participants undergoing assisted reproduction technology or trying to become pregnant. For example, Pam (25–34) shared, “I guess that is the real challenge for anyone with fertility issues who are going through treatment is that you’re just so limited in what you can take and how you can treat it.” Consequently, participants described that a range of treatment options and modalities should be available to manage endometriosis symptoms.

Non-contraceptive treatment options for pain management were reported as an urgent need. This was due to the significant side effects of hormonal contraceptive treatment experienced by participants as well as their inaccessibility to patients who are trying to conceive. As Sakura (25–34) described, “Contraception is a bloody Band-Aid. (…) I’ve been on so many different pills. I’ve had the Mirena. They all drive me absolutely up the wall.” She went on to explain that the effects of coming off of contraception led to “flare ups,” difficulty with bowel movements, the need for “painkillers,” or a visit to the emergency department (E.D.).


*“My husband and I want to try and have children, but it worries you coming off the pill, because you know anytime you have a period, you usually end up flaring and winding up in ED or you spend three days on so many painkillers that you then can’t go to the toilet and it just ends up being a cyclical, terrible time. I think that there needs to be other ways and better first line treatments than just whacking every person with endometriosis on some form of contraception.”*
(Sakura, 25–34)

For participants who were not trying to conceive, the use of medicinal cannabis was seen as a “natural” way to ameliorate symptoms that stood in contrast to biomedical treatments that could make “you feel terrible.” As explained by Lena:


*“I really wish that it [medicinal cannabis] was available, because I’m sick to death of taking medication that makes you feel terrible, that is highly processed and highly, highly chemical-base substances stripping your body, where at least cannabis comes from a natural derived product.”*
(Lena, 18–24)

Medicinal cannabis was a popular potential treatment choice, with 15 out of 30 participants being interested in research on medicinal cannabis for endometriosis pain ma-nagement. Nevertheless, there were concerns regarding the use of cannabis due to legality issues, which might result in “losing jobs.” Safety in relation to the “effects of cannabis on the developing brain,” its “psychoactive effects,” and the potential for “long-term effects” were also raised, highlighting the need for research to move beyond pain relief effectiveness and explore issues of medicinal cannabis and long-term safety.

Nineteen out of 30 participants reported using non-medical treatment for addressing pain such as diet, heat packs, cannabidiol (CBD) oil, a transcutaneous electrical nerve sti-mulation (TENS) machine, meditation, yoga, Pilates, rest, magnesium, turmeric, fish oil, Epsom salt or chloride baths, and other forms of complementary and alternative medicine (CAM). For instance, Diana (18–24) said, “I love my heat packs, but acupuncture is pro-bably the really big one for me that’s non-medicated.” CAM treatment options mentioned by the participants included acupuncture, Chinese herbs, cupping, chiropractic, (pelvic) physiotherapy, and massage therapy. Participants felt that these should be offered as part of a holistic treatment that includes “nutrition, massage, acupuncture [and] other types of therapies” (Wendy, ≥35).

A holistic treatment would require the collaboration of various health-care providers such as acupuncturists, osteopaths, physiotherapists, and nutritionists. “So, the acupuncture, the osteopath, physiotherapist help you. All of those people can [come] together and work out a plan” (Gloria, ≥35). Accordingly, four participants expressed a desire for a clear treatment plan that people with endometriosis can follow. As Kate (26–35) said, “I wish [the doctor] had just put a clear treatment plan in place five years ago.”

Eight out of 30 participants spoke of the psychological impact of endometriosis as a “struggle,” describing their feelings as “depressed,” “more anxious,” “angry, so angry,” “desperate,” and characterising their endometriosis journey as an “emotional rollercoaster,” without being able to do much about it: “so much time just lying in bed crying” (Genevieve, 18–24). Accordingly, another research priority must focus on the psychological implications of endometriosis.

Four participants indicated interest in research on the “causes” of endometriosis to develop a cure, since pain management was considered to be a “band-aid” (Naomi, ≥35). As Caroline stated, more research is needed “into what actually creates endometriosis at a certain degree, so what effects that, what makes it grow” (Caroline, ≥35). These participants reported that knowing what causes endometriosis is likely to lead to treatment options to prevent endometriosis from progressing. For Gina, knowing whether endometriosis is hereditary would be a useful method for determining treatment options:


*“I’d really like to find out what exactly causes it, although that’s not gonna be easy, and just a genetic component, whether or not it gets passed down to your children and your grandchildren and if there is anything, we can do.”*
(Gina, 25–34)

Other minority views included the need for research into the immunological causes of endometriosis, whether endometriosis is an autoimmune disease, the impact on fertility of subgroups, and a deeper understanding of the effects of medications (like antibiotics) on endometriosis symptoms. Table 2 summarises the research priorities based on the unmet needs of participants.

### 3.2. Motivators and Barriers to Participation in Endometriosis Research

The participants’ accounts indicated significant interest in participating in endometriosis research. Wendy (≥35) stated, “I’m at a desperate point at the moment so I’m trying anything and everything. So, I would try whatever you [the researchers] got.” According to Mindy (25–34), “I think a lot of endo women would probably jump at the chance to be involved in anything that could help.”

Eight out of 30 participants described that the motivation for participation in research is based on the expectation that the treatment provided may alleviate symptoms, making everyday life easier. As Rosa (25–34) said, “if it’s gonna help day-to-day living, it’s worth it trying anything.” However, four participants also felt motivated by a need to help others, as expressed by Lucy (≥35): “You’re doing it for the altruistic reason.” The altruistic reason to participate in endometriosis research was mostly highlighted in the group aged 35 and above. Caroline (≥35) explained, “You’re doing it, so that you can help others and so that the [time to] diagnosis is not so long, so you’re willing to help. You don’t want a reward, don’t need a reward. It’s that you’re helping.”

The motivation to help others through early detection became even more important when it came to the participants’ own daughters:


*“I personally would go through heaven and hell to find something. I would take that placebo drug. I hate yoga. I would do yoga, I’ll go jogging for an hour every single day, if it meant that in the future, my daughter, there’s something there for her.”*
(Angela, 18–24)

Three other participants discussed that participating in research depends on the research topic. Pam (25–34) indicated that the research would need to align with her own individual situation: “I would commit time if I had a goal or a set thing that I was trying to achieve, then I would attempt to commit time to it.”

Since research participation is often driven by the expectation to relieve symptoms, the research topic was deemed as relevant especially if the treatment does not fit with the individual’s aims, beliefs, or former experiences. Pam (25–34) explained, “Yeah, what they’re actually studying. I chose not to go in an endometriosis study because it was just a different type of contraception.”

In research, the effectiveness of an intervention must be tracked using measurements such as pain scores, mental health levels, and blood parameters. According to 12 out of 30 participants, access to user-friendly tracking was seen as key to participation. The partici-pants expressed that a reminder should be sent when tracking is to be performed. If the tracking is feasible and quick to complete, tracking every 2–3 days to daily was described as possible.


*“If it was in a handy little app and it sent you a notification every day and it was super easy, tap-tap-tap, that would be fine. But if you’ve got to remember and log-in somewhere that’s clunky, then likely that you’d forget or not feel like it.”*
(Carolyn, 18–24)

Six participants stated that participation in research is considered more likely when invasive tests, such as blood collection, occur at an accessible time and location. Blood collection was described as challenging during the first few days of the menstrual cycle, as it is difficult for consumers to predict their pain symptoms. Flexible timing and thus a participant’s ability to travel is therefore essential. As Jan (25–34) said, “If it was in the first couple of days, then I probably need to organise for someone to drive me depending on where it was just because I can’t really manage alone.” Five participants expressed interest in receiving the results of blood tests collected as part of the study, or, as Naomi (≥35) explained, “I’d be very interested to get them [the results of the blood test] if I was having them regularly.”

Being offered an incentive for research participation such as a gift card that would offset the cost of participation was discussed differently in the various age groups. In the 18- to 24-year-old group, five participants agreed that it would be beneficial to have an incentive for research participation. The request for an incentive parallels concern about problematic high medication costs. It was expressed that a financial incentive commensurate with the effort would be useful. “It depends how invasive it is. The more invasive, I think the more incentive there needs to be to get them to stick around.” (Melissa, 18–24)

In contrast, in the two older participant groups only two participants indicated an interest in being offered an incentive, whereas the majority (10 participants) explicitly stated that an incentive to participate in a research project was not important to them.

Finally, participants stated that they access information about studies from endometriosis support groups, social media such as Facebook or Instagram, posts on university websites, or, in rural areas, at health centres. It is likely that a widespread dissemination of information about future research supports participant numbers. Overall, although the motivation for taking part in research is generally high, there are individual limiting factors that are set within the social context of an individual’s life.


*“I think it [the participation in a research project] would really depend like for me, it would depend on what’s going on in my life and what the study is measuring for me, it would be dependent on whether I want to be a part of it but I’m pretty likely to.”*
(Pam, 25–34)

Four main barriers to participation were identified across the participant accounts. The most significant barrier was the requested time commitment for research participation. It was suggested that participation needs to fit into the daily schedule of people with endometriosis or, as Gina (25–34) said, “around working hours,” as she stated that she does not “really wanna take any time off work.” Flexibility was stated as a key component for research participation. It should be possible to choose the time of day as well as the days of the week. Jan (25–34) explained, “It would depend, for me, what part of my cycle I’m in, that’s a big factor for me. At the moment, I go to the pool two to three times a week but not during my period.”

Participants’ responses varied regarding the time they could commit to a research study. A number of participants (10/30) stated two to four times per week for half an hour to an hour, whereas others stated either less (a maximum of half an hour to an hour per week, as stated by four participants) or more (“as much time as it needs to make me feel better” (Wendy, ≥35)). Individual responses included that participation depends on their work situation and the potential benefits of participation.


*“If you’re being really super practical about it, I would think, ‘How much is it helping me the thing you’re asking me to do’, and also what’s the compensation. If it’s gonna be something that’s really time consuming, I might think, ‘well, I’m getting a gift card that’s paying my groceries a week’, so I’ll be able to do that. But if there’s nothing in it for me and it might be a placebo drug I’m on, it’s just gonna feel like a waste of time. I want to be benevolent and help the future of research but I’m a selfish human.”*
(Diana, 18–24)

If research-related tasks can be performed at home, two participants reported being more inclined to do so more frequently than if they have to travel to complete the task. Pam indicated that the location would make a difference, saying, “I think anything that you can do at home, you’re more likely—well, I would be way more likely there to do more frequently.” (Pam, 25–34)

For Sakura, if she must travel to participate, proximity is key: “If I need to go to Melbourne frequently with that, it’s just a lot to have to do. But if it was available locally, then there wouldn’t be a problem at all” (Sakura, 25–34).

As another barrier to participation in endometriosis studies, safety concerns were raised by eight participants: If there is a risk that participation in a research project will increase pain symptoms, participants reported not wanting to participate. Safety with current medication must be ensured. Gina (25–34) stated, “My only other thing would be to see if we could link it with our current medical things just so that they’re aware of like what’s happening and so that they can be built into our plan of care.”

Individual incompatibilities must be considered when prescribing medication in a research setting or, as Lena (18–24) stated, “making sure that they didn’t have the herbal supplement, didn’t have any contraindications with any of the drugs that the person was taking, 100%, I’d be on board.”

In research on medicinal cannabis, safety becomes even more important in terms of long-term effects, as stated above, and legal issues. Lena explained:


*“I know that I’d lose my registration, so making sure that it was legal. When I’m drug-tested at work, making sure that I either had some legal documentation to say, ‘I’d take this for chronic health condition’, or whether it didn’t show up in that drug test, I know that sounds hilarious, but just having some form of back-up or even for instance, if we were to go onto this study with medicinal cannabis, having a letter from the university or from medical practitioner saying, ‘I am currently doing a clinical trial for this, this and this reason’, the clinical trial is on medicinal cannabis, whatever reason. Just making sure my bum was covered at work is a big thing for me.”*
(Lena, 18–24)

A minority view stated by four participants who were trying to conceive was that they did not want to participate in research projects. Overall, issues of conception played a particularly important role in the 25–34-year-old group. The issue was raised both in the context of wanting to conduct research on non-contraceptive treatment options to reduce pain symptoms and as a concern about participating in research projects, particularly but not exclusively projects exploring medicinal cannabis. Table 3 provides an overview of the motivators and barriers that shape participation in endometriosis research.

## 4. Discussion

Our study found that Australian people with endometriosis across a broad age range had several research priorities that were unmet at the time. This included the importance of a non-invasive diagnosis to help shorten diagnostic delay, a wider range of non-hormonal and non-surgical treatment options, and a focus on understanding the cause and thus developing a cure for endometriosis. Despite the strong interest in participating in research projects, motivated in part by anticipated symptom relief and a desire to help others, several barriers were found, including a high time commitment and cost associated with participating in a research project.

The findings of this study support previous research that highlights the high priority for developing new methods or improving existing tools for non-invasive diagnosis [10,12,17,25] and research into the causes of endometriosis for the development of a cure [10,17]. In line with previous work [26], there was significant dissatisfaction with current treatments for endometriosis, including hormonal treatment and surgery, with our participants wanting an urgent shift toward new, non-hormonal treatment options and a more holistic approach to treatment. The latter has been previously called for by consu-mers, particularly to address the numerous comorbidities associated with endometriosis that affect quality of life [12]. Our findings on unmet needs leading to consumer-relevant research topics differ from previous publications in one major respect: Previous research driven by researchers has largely focused on improving existing tools or treatments [10,17], whereas the consumers in our study desired a focus on new tools and treatment options that can improve their overall well-being.

This difference is unsurprising, given that previous research has found discrepancies between research priorities stated by patients or close family members compared to those formulated by healthcare providers and scientists [21], as demonstrated in the current study. Since dominant personalities may influence the outcome of face-to-face consensus groups [27], the presence of health-care professionals in priority-setting partnerships may lead to endometriosis patients’ holding back thoughts that can be more freely pronounced in a setting without people who are presenting the current system. When people with endometriosis express their views without being influenced by the presence of health-care practitioners, discrepancies become clear.

Firstly, according to the findings of this study, there is a need for research to develop a non-contraceptive first-line treatment so that seeking pregnancy does not leave endometriosis patients untreated. Secondly, since participants indicated interest in research on the use of medicinal cannabis and a holistic treatment including coordinated use of CAM therapies, research is needed on effective CAM to enhance multidisciplinary treatment plans. Since CAM treatment costs are not yet covered by insurance in Australia, future research needs to investigate the effectiveness of non-medical and CAM treatments so that evidence-based decisions about coverage of non-medical treatment options can be made.

Based on the research priority differences between the various age groups, two areas of relevance to research design emerged: Participants’ statements in the 18–24-year-old group indicated an issue with high treatment costs and an interest in an incentive for research participation to balance travel costs. Consequently, low-cost treatment options, especially for younger Australians with endometriosis, are needed, and an incentive for research participation with this age group should be part of research planning. For the middle-aged group, for those who are trying to get pregnant, conception is key to their decision-making. This is in line with formerly published research recommendations [10,17]. Researchers need to consider that attrition due to reproduction is likely in that age group.

Considering patients’ needs when designing a research project involving people with endometriosis is assumed to improve research quality. Involving consumers in research design has beneficial effects and leads to higher quality and more clinical relevance because of the unique perspective that consumers can bring to a research project [19]. In research, co-designs are used to meet the needs of those affected [28], foster acceptance by target users [29], offer a more sustainable and effective translation approach into clinical practice, increase the effectiveness of the intervention [30], and improve the quality and appropriateness of study design [31]. Tay et al., who reviewed co-design practices in diet and nutrition research, reported that a high percentage (75%) of studies that showed posi-tive outcomes were those that involved end-users in prototype testing, followed by those that assessed user needs to inform intervention focus (67%) and those that involved end-users in pilot testing (67%) [32].

To our knowledge, no research to date has addressed endometriosis patients’ motivators and barriers to participation in research. Taking patients’ needs into account is specifically important in endometriosis research, as high dropout rates have been reported in the literature: Bergqvist et al. reported that 40 out of 48 patients did not complete the 18-month-long follow-up period [33]. They stated that the highest dropout numbers in the placebo groups were due to insufficient efficacy. Wright and Redwine stated high dropout rates not only in research but also in treatment regimens due to intolerable side effects [34]. Kuivasaari et al. reported a dropout rate of 52.2% by people with stage III/IV endometriosis compared to 38.7% of participants with milder endometriosis in an observational study [35]. These examples indicate that dropouts not only occur in placebo-groups but are also based on an unsatisfying treatment experience and increases in patients with a higher stage of endometriosis—indicating that patients with a higher burden have more problems fulfilling the requested tasks of the research.

As a consequence, future research can take the patients’ needs defined in this research project into account to make participation more likely and minimise dropout rates. Therefore, it is essential to co-design and co-produce endometriosis research across the whole research cycle with consumers.

### Strengths and Limitations

A main strength of this project is that the focus groups were run as discussions, where participants were able to respond to each other and get involved in an exchange. These discussions were driven more by participants than researchers and enabled participants to express experiences and opinions freely.

Another strength is that the recruitment included a wide range of ages and geographical locations, a key consideration in a country as vast as Australia, as the experiences and views from participants differed by age, and those in rural and remote areas have significant challenges that may not be present in urban areas, where treatment options are more plentiful.

By providing insight into the unmet needs of patients, future research projects can focus on non-invasive diagnostic tools and effectiveness research into non-hormonal treatment strategies that consider the patient as a whole rather than focusing on specific symptoms. Furthermore, given the drivers and barriers to study participation presented, it is critical to co-design research projects consistent with patient needs. Since placebo-controlled research designs are highly problematic in endometriosis due to noncompliance, observational study designs may be considered in certain cases.

Nonetheless, there are limitations that must be acknowledged. Given that the recruitment included Australians only, the findings may not be generalizable to other parts of the world. The challenges of research participation by rural and remote people may not be as pronounced in more densely populated countries. However, it is likely that people around the world who live in rural areas with long distances to major cities face the same problem. Similarly, research priorities are likely to be the same for Australian endometriosis consumers and worldwide.

At the same time, recruitment via social media allowed us to ensure a broad demographic; however, it must be noted that recruitment via social media tends to include those with more severe symptoms and worse quality of life [36]. Therefore, those in the community more generally may have less severe symptoms, and this may, in turn, change their priorities. However, given the consistency between our findings and international findings, the impact of this is likely to be minimal.

## 5. Conclusions

People with endometriosis in Australia reported several research priorities, including improving non-invasive methods of diagnosis and providing more options for effective non-hormonal treatment strategies, especially those that were viewed as more holistic, such as acupuncture and herbal medicine. Medicinal cannabis was a popular target for more research. Our participants reported a willingness to engage in future research, with the main driver being the expected relief of symptoms. Considering the significant and often unpredictable symptoms experienced by people with endometriosis, it is vital that care is taken in the development of clinical trials to avoid undue burden on participants. In particular, this includes requiring flexible time commitments and tasks that can be completed at home or locally, as barriers such as travel must be considered for research to be viable. Consumer-relevant research and designs that fit the specific needs of participants can be ensured by co-designing and co-producing endometriosis research with consu-mers. Future research should aim to enable endometriosis management that meets the needs of patients and focuses on the well-being of the whole person.

## Figures and Tables

**Table 1 medicina-59-01655-t001:** Characteristics of participants (*n* = 30).

Characteristics		*n* (%)
Age (years)	18–24	7 (23)
	25–34	14 (47)
	≥35	9 (30)
Employment status	Full time (≥35 h/week)	18 (60)
	Part time (<35 h/week)	7 (23)
	Self-employed	2 (7)
	Studying and working part time	1 (3)
	Not employed, looking for work	1 (3)
	Disabled, not able to work	1 (3)
Relationship status	Married	19 (63)
	Living together, not married	6 (20)
	Single, never married	5 (17)
Children	No children	25 (83)
	1	1 (3)
	2	3 (10)
	3	1 (3)
Ethnicity	Caucasian	24 (80)
	Aboriginal or Torres Strait Islander	1 (3)
	European (Western or Northern)	2 (7)
	Southeast Asian	1 (3)
	Mixed (Caucasian and others)	2 (7)
Highest education/degree	Secondary school	5 (17)
	Technical college or other college	9 (30)
	University undergraduate	12 (40)
	University diploma	1 (3)
	University postgraduate (master’s or Ph.D. *)	3 (10)

* Ph.D.: Doctor of Philosophy.

**Table 2 medicina-59-01655-t002:** Research priorities from the perspective of endometriosis patients in Australia. * CAM: Complementary and alternative medicine.

Research Priorities	
Diagnosis	Development of a non-invasive diagnostic tool
Treatment: endometriosis treatment options to manage symptoms	Development of a non-contraceptive first-line treatment
	Effectiveness and safety of medicinal cannabis for endometriosis for legalisation in Australia
	Research on a holistic treatment approach including CAM * therapies
	Treatment options for emotional distress
Cure	Research on causes of endometriosis for cure development

**Table 3 medicina-59-01655-t003:** Motivators and barriers to participation in endometriosis research in Australia.

Motivators	Barriers
Helping yourself: Expected relief of symptoms Help for daily life	Time commitments that do not fit into the schedule; possible time commitments: 1–4 times a week for 30–60 min, flexible
Helping others: Shortening the time to diagnosis	Localisation: At home preferably and more frequently, otherwise locally
Research topic: according to personal aims, believes, and former experiences	Safety: must be ensured Anticipation of negative impact on pain symptoms
Tracking: user-friendly, reminder, fast to complete	Trying to conceive
Blood test: close by, flexible, obtain the results of the blood tests	
Financial incentive: not required for all patient groups; depending on cost and effort	
Dissemination: through support groups, social media, university websites, and health centres	

## Data Availability

The anonymised transcripts analysed for the current publication are available from the corresponding author on reasonable request.

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
