# Peer review of "Unmet Needs of Australians in Endometriosis Research: A Qualitative Study of Research Priorities, Drivers, and Barriers to Participation in People with Endometriosis"

_medicina, 2023, doi:10.3390/medicina59091655_

Round 1

Reviewer 1 Report

I appreciate the opportunity to review the manuscript entitled “Unmet needs of Australians in endometriosis research: a qualitative study of research priorities, drivers and barriers to participation in people with endometriosis” submitted in journal Medicina.  

The authors present the results about research priorities and determinants of endometriosis research from the perspective of people with endometriosis in Australia.

Reviewer Comments:

1.      Please prepare list of abbreviations used in manuscript.

2.       Please extand the discussion with the more results obtained from other authors.  What are the findings of other authors in the research about the research priorities, drivers and barriers to participation in people with endometriosis

3. What are strengths and limitations of yours study­­? Consider the relevance of yours study for the endometriosis research.

Taking into account the significance of the presented topic and results in above mentioned manuscript, my opinion is that this submission meet the criteria to be published in journal Medicina after major revisions and inclusion of the paragraphs I suggested.

Author Response

Dear colleague,

Thank you very much for reviewing our manuscript and for your helpful comments and feedback.

We addressed all issues as follows:

Please prepare list of abbreviations used in manuscript

We added a list of abbreviations at the end of the manuscript.

“List of abbreviations

CAM     Complementary and alternative medicine

CBD      Cannabidiol

ED         Emergency department

TENS    transcutaneous electrical nerve stimulation”

Lines 549-553

Please extend the discussion with the more results obtained from other authors. What are the findings of other authors in the research about the research priorities, drivers and barriers to participation in people with endometriosis?

The main information about the research priorities formulated in the past can be found in the introduction. We believe that this type of information is better placed at the beginning of our manuscript rather than in the discussion. We have added a section to the discussion that provides an overview of former research related to the findings of our study. Other aspects of previous research are discussed in the following sections.

“The findings of this study support previous research that highlight the high priority for developing new methods, or improving existing tools, for non-invasive diagnosis [10,12,17,25] and research into the causes of endometriosis for the development of a cure [10,17]. In line with previous work [26], there was significant dissatisfaction with current treatments for endometriosis, including hormonal treatment and surgery, with our participants wanting an urgent shift toward new, non-hormonal treatment options and a more holistic approach to treatment. The latter has been previously called for by consumers, particularly to address the numerous comorbidities associated with endometriosis that affect quality of life [12]. Our findings on unmet needs leading to consumer-relevant research topics differ from previous publications in one major respect; previous research driven by researchers has largely focused on improving existing tools or treatments [10,17], while the consumers in our study desired a focus on new tools and treatment options that can improve their overall well-being.”

Unfortunately, there is no previous research on drivers and barriers to participation by people with endometriosis. We added a sentence to the discussion.

“To our knowledge no research to date has addressed endometriosis patients’ motivators and barriers to participation in research.“

Lines 435-447

Lines 487-488

What are strengths and limitations of yours study­­? Consider the relevance of yours study for the endometriosis research.

The strength and limitations were presented in section 4.1. We added a section on the relevance for future endometriosis research.

“By providing insight into unmet needs of patients, future research projects can focus on non-invasive diagnostic tools and effectiveness research into non-hormonal treatment strategies that consider the patient as a whole, rather than focusing on specific symptoms. Furthermore, given the drivers and barriers to study participation presented, it is critical to co-design research projects consistent with patient needs. Since placebo-controlled research designs are highly problematic in endometriosis due to noncompliance, observational study designs may be considered in certain cases.”

Lines 514-520

With best regards,

Nora Giese

Reviewer 2 Report

Dear Authors,

I have appreciated a lot the topic of your manuscript since endometriosis still represents a current disease, often difficult to manage. Chronic pain associated to endometriosis has a negative impact on QoL of patients affected, thus an early diagnosis and treatment could reduce the burden related to it.

Thus I think your manuscript could be useful to enrich literature with an actual point of view on this topic especially from patients.

What is the main question addressed by the research?

Improvement in management of endometriosis y promoting patients' access to studies and centres and improvement  of a tailored research on this topic
 Do you consider the topic original or relevant in the field? Does it
address a specific gap in the field?  According to my opinion the topic is original
What does it add to the subject area compared with other published
material?  It adds a more recent point of view on this burden disease to address the research in a more tailored manner even if not exclusively based on hormonal or surgical treatment.    Are the references appropriate?  Yes, they are
Please include any additional comments on the tables and figures. Tables are clear and well-structured. No figures are present in the manuscript.   Minor English revision is needed. 

Author Response

Dear colleague,

Thank you very much for reviewing our manuscript and for your helpful comments and feedback.

We addressed all issues as follows:

Presentation of results can be improved.

The results are now displayed more clearly to provide a clear visual distinction between citations and text: Quotations indented, italicized, blank lines before and after quotations.

Section 3

Minor English revision is needed

We have revised the manuscript with regard to the use of English.

Full text

With best regards,

Nora Giese

Reviewer 3 Report

Dear Authors

Please clarify your main conclusion of this research.

What would you want to plan in terms of endometriosis managment after this research?

Manuscript need english editing

Author Response

Dear colleague,

Thank you very much for reviewing our manuscript and for your helpful comments and feedback.

We addressed all issues as follows:

Moderate editing of English language required.

We have revised the manuscript with regard to the use of English

Full text

Presentation of results must be improved

The results are now displayed more clearly to provide a clear visual distinction between citations and text: Quotations indented, italicized, blank lines before and after quotations.

Section 3

Please clarify your main conclusion of this research.

Revision of conclusion.

“People with endometriosis in Australia reported several research priorities including improving non-invasive methods of diagnosis and providing more options for effective non-hormonal treatment strategies, especially those that were viewed as more holistic such as acupuncture and herbal medicine. Medicinal cannabis was a popular target for more research. Our participants reported a willingness to engage in future research, with the main driver being the expected relief of symptoms. Considering the significant and often unpredictable symptoms experienced by people with endometriosis, it is vital that care is taken in the development of clinical trials to avoid undue burden on participants. In particular, this includes requiring flexible time commitments and tasks that can be completed at home or locally, as barriers such as travel must be considered, for research to be viable. Consumer-relevant research and designs that fit the specific needs of participants can be ensured by co-designing and co-producing endometriosis-research with consumers. Future research should aim to enable endometriosis management that meets the needs of patients and focuses on the well-being of the whole person.”

Section 5

What would you want to plan in terms of endometriosis management after this research?

Endometriosis management was not our research topic - we focused on research priorities rather than how to manage treatment. However, based on our findings, we were able to include a sentence on endometriosis management.

“Future research should aim to enable endometriosis management that meets the needs of patients and focuses on the well-being of the whole person.”

Lines 547-548

With best regards,

Nora Giese

Round 2

Reviewer 1 Report

My opinion is that this submission meets the criteria to be published in journal Medicina. The authors accepted suggested comments and prepare the manuscript according them.